# Purinergic Signaling in Pancreas—From Physiology to Therapeutic Strategies in Pancreatic Cancer

**DOI:** 10.3390/ijms21228781

**Published:** 2020-11-20

**Authors:** Ivana Novak, Haoran Yu, Lara Magni, Ganga Deshar

**Affiliations:** Section for Cell Biology and Physiology, Department of Biology, University of Copenhagen, 2100 Copenhagen Ø, Denmark; haoran.yu@bio.ku.dk (H.Y.); lara.magni@bio.ku.dk (L.M.); ganga.deshar@bio.ku.dk (G.D.)

**Keywords:** pancreatic cancer, PDAC, pancreatic stellate cells, PSC, fibrosis, immune cells, inflammation, pancreatitis, ion channels, immunotherapy

## Abstract

The purinergic signaling has an important role in regulating pancreatic exocrine secretion. The exocrine pancreas is also a site of one of the most serious cancer forms, the pancreatic ductal adenocarcinoma (PDAC). Here, we explore how the network of purinergic and adenosine receptors, as well as ecto-nucleotidases regulate normal pancreatic cells and various cells within the pancreatic tumor microenvironment. In particular, we focus on the P2X7 receptor, P2Y_2_ and P2Y_12_ receptors, as well as A_2_ receptors and ecto-nucleotidases CD39 and CD73. Recent studies indicate that targeting one or more of these candidates could present new therapeutic approaches to treat pancreatic cancer. In pancreatic cancer, as much as possible of normal pancreatic function should be preserved, and therefore physiology of purinergic signaling in pancreas needs to be considered.

## 1. Introduction

Pancreatic cancer is one of the deadliest forms of cancer, and new insights into the biology of this cancer and therapeutic approaches are needed. There is a spur of research in genetics and origin of pancreatic cancer, and a recent agenda is the exploration of the cancer tumor microenvironment that includes a variety of resident and recruited cells, cell–cell crosstalk, and effects of molecular/chemical components on these. From research on other cancer forms, it is known that the latter includes nucleotides/sides interacting with purinergic receptors, various modulating enzymes and transporters that all jointly participate in purinergic signaling, as summarized in several reviews referred to below. Thus, in the tumor microenvironment (TME), necrotic cells, metabolically active cancer cells and stromal cells, and factors such as hypoxia and mechanical stress can cause release of ATP that is more substantial than in the physiological state. It is predicted that extracellular ATP forms a landscape/geographical profile across the tumor, from necrotic core with high ATP to peripheral halo with lower ATP [1]. Extracellular ATP has a palette of positive and negative effects, depending on target cells and their ATP-binding receptors, which range from boosting anti-tumor immune response to cancer cell death or proliferation, fibrosis, impaired vascular permeability, and metastatic development [2,3,4,5]. Furthermore, tumor hypoxia stimulates expression of ATP-hydrolyzing enzymes (CD39 and CD73) with a net result in increased level of adenosine, which is immunosuppressive and affecting cancer cells, as well as contributing to resistance to chemotherapy [6,7,8,9]. Purinergic signaling thus modulates a multitude of steps in tumor progression, spanning from dysregulated cell growth and metabolism to hypoxia, angiogenesis, acidic TME, and lastly to extravasation and metastatic dissemination. 

The plasma membrane receptors expressed on various cells in the TME can be ATP-stimulated ion channels/receptors from the P2X family (P2X1–P2X7), and G-protein coupled P2Y receptors belonging to two structurally different families (with preferred agonist given in brackets): mostly G_q_ coupled receptors P2Y_1_ (ADP), P2Y_2_ (ATP/UTP), P2Y_4_ (UTP), P2Y_6_ (UDP), and G_q_ and G_s_ coupled P2Y_11_ (ATP); and mostly G_i_ coupled receptors P2Y_12_ (ADP), P2Y_13_ (ADP), P2Y_14_ (UDP/UDP-glucose) [10,11,12]. The receptors stimulated by adenosine (P1 receptors) are G_i/o_-coupled A_1_ and A_3_ receptors, and G_s_-coupled A_2A_ and A_2B_ receptors [7,13]. Nucleotide/side-modifying enzymes include ecto-ATP/ADPases (ecto-nucleoside triphosphate diphophophydrolases, eNTPDases, CD39 family), 5′-nucleotidases (CD73), ecto-kinases, and pyrophosphates [14,15,16]. A number of ion channels/transporters, such as pannexin-1 (Panx-1), and vesicular nucleotide transporters (VNUT), coded by *SLC17A9*, are involved in ATP release [17,18,19]. Adenosine is thought not to be released by cells, but to large extent originates from nucleotide hydrolysis and then is taken up into cells via nucleoside transporters, which are in fact targets for delivery of cytotoxic chemotherapeutic nucleoside drugs [16]. Expression and function of the above mentioned purinergic players might vary depending on cancer form, and there are a number of excellent reviews on this topic in general [4,6,7,16]. However, there is no comprehensive/detailed account on the role of purinergic signaling in pancreatic cancer, and this is the main purpose of our review.

Purinergic signaling is important in regulation of pancreatic exocrine function as detailed below. Pancreatic ductal adenocarcinoma (PDAC), which is the most common form of pancreatic cancer, originates most likely from exocrine cells. Some of the special features of pancreatic tumor are that it is highly fibrotic with high turgor pressure, hypo-vascularity, often with perineural invasion and with high immunosuppressive milieu—implicating complex cellular and physical/chemical interaction within TME, which need to be understood in order to provide effective therapy. In this review, we will briefly summarize main features of pancreatic cancer and its TME, and then we will take the purinergic signaling in exocrine pancreas as a starting point to understand the role of purinergic signaling in pancreatic cancer. Purinergic signaling in pancreatic cancer is a relatively new research field and, in some instances, our knowledge needs to be supplemented with references to other cancer forms. Nevertheless, some promising results warrant further investigations and understanding of purinergic signaling proteins/receptors as novel therapeutic approaches to treat pancreatic cancer.

## 2. Pancreatic Cancer and Its Tumor Microenvironment

The most common form of pancreatic cancer is the exocrine-derived pancreatic ductal adenocarcinoma (PDAC). It is one of the leading causes of cancer-related deaths with an average 5-year overall survival rate of about 9% [20]. Pancreas has a great redundancy in exocrine and endocrine function, so that a tumor within pancreas may remain undetected, unless it develops at the head of pancreas obstructing bile outflow through the common bile duct. Due to lack of visible symptoms, as well as specific biomarkers, patients are usually diagnosed at an advanced stage, which limits options of surgical intervention, as patients already have aggressive metastatic disease that does not respond well to cancer treatment [21]. Two other characteristics of PDAC/pancreatic cancer make it difficult to treat. First, PDAC is a solid tumor, rich in stromal cells and exhibiting marked fibrosis, stiffness, hypoxia, acidosis and poor vascularization, which makes the drug delivery at the primary tumor site difficult [21,22,23]. Second, pancreatic tumor harbors immune cells with immunosuppressive phenotype [21,24,25]. Apart from genetics, life-style factors, and age, some of the well-recognized risk factors for pancreatic cancer are diseases with inflammatory components such as chronic pancreatitis, obesity, and diabetes [21,26]. Main mutations that drive PDAC development are activating mutations in *KRAS*, and inactivating mutations in *TP53*, *CDKN2A*, and *SMAD4*, but there is a great number of evolving mutations [27,28,29]. Recent years of intense research brought excellent knowledge about the epidemiology and genetics of the disease [21,27,30]. Nevertheless, our understanding of the cellular and molecular mechanisms that give rise to PDAC and metastases is still inadequate and limits the development of novel therapeutic approaches. In particular, focus on the cellular and molecular mechanisms in the TME, as well as integrating analysis of histological, genomic and transcriptional and particularly functional features of both tumor and metastatic sites, may bring new strategies [30,31]. 

The cellular origin of pancreatic cancer cells is still debated and candidates include exocrine duct cells that loose epithelial polarity, become neoplastic and go through pre-cancerous pancreatic intraepithelial neoplasias (PanIN) to cancer [32,33]. PanINs can also arise from other ductal neoplasias, such as mucinous cystic neoplasm or intraductal papillary mucinous neoplasm. Another path involves acinar re-programing and trans-differentiation leading to acini-duct metaplasias (ADM) [34,35,36]. Both PanIN and ADM processes may be in parallel or in tandem and there might be differences between rodents and humans [37,38]. 

Among the cancer cells, less than 1% of the population are cancer stem cells (CSCs). This pool of cells possesses self-renewal, multi-lineage differentiation, and tumor initiating capacity. They promote tumor growth and progression through many mechanisms, including modification of adjacent stromal cells, interaction with immune cells and evasion of conventional therapies. There are several sub-populations expressing specific markers for stemness, metastatic potential, chemoresistance, immunoevasion, and links/receptors for epithelial to mesenchymal transition (EMT). Understanding the relationship between CSCs and TME components is important to improve the knowledge of the CSC biology [39,40].

Apart from cancer cells, pancreatic TME includes diverse stromal cells including fibroblasts, pancreatic stellate cells (PSCs), immune cells, nerves, and blood vessels. PDAC is characterized by extensive desmoplastic reaction mediated by heterogenous population of cancer-associated fibroblasts (CAFs) [41], including the most studied PSCs. PSCs have the ability to repair, respond to stress, and induce fibrosis; and thus are regarded as multi-functional cells with both a homeostatic role in pancreas physiology and in pancreatic diseases, such as cystic fibrosis, pancreatitis, and PDAC [42]. In healthy pancreas fibroblasts/PSCs comprise 4–7% of all pancreatic cells, but in PDAC they can form more than 80% of the tumor [23,43]. PSCs can be activated by different pathogenic factors including pancreas injury, ethanol, endotoxin, growth factors, oxidant stress, cytokines [43,44], and as we discuss below—by extracellular ATP. Activated PSCs play a central role in PDAC as they secrete many factors including cytokines, chemokines and extracellular matrix (ECM) proteins. PSCs can induce EMT, activate cancer cells, suppress immune cells, contribute to stiffness and impermeability of the tumor, invasion and dissemination of cancer [21,42]. 

The PDAC microenvironment includes an inflammatory cell population that is unbalanced and tends to be of an immunosuppressive phenotype. There is a prevalence of myeloid derived suppressor cells (MDSCs), M2 polarized macrophages, and regulatory T cells (T_reg_). In contrast, there is a depletion of CD4^+^ and CD8^+^ T lymphocytes, M1 macrophages and dendritic cells [22,25]. There are several mechanisms that contribute to immunosuppression in pancreatic cancer and involve cell–cell crosstalk and components secreted into TME. For example: a number of cells secrete immunosuppressive transforming growth factor TGF−β; PSCs secrete CXC chemokine ligand CXCL12 that reduces the migration of CD8^+^ T cells, and PSCs also secrete galectin-1 that promotes immunosuppression by inducing T cell apoptosis and Th2 cytokine secretion [21,22,45,46]. Activated PCS can also produce signaling molecules to recruit other cells like T_regs_, MDSCs and tumor-associated macrophages (TAMs) into the cancer microenvironment and contribute to pancreatic cancer progression, inflammation as well as metastasis [22,47]. Probably, one of the most important mechanisms in cancer immune-evasion involves programmed cell death pathway, where cancer cells and PSCs express programmed cells death protein-1 ligand (PD-L1) that binds to PD-1 receptor on T cells (and other immune cells) and suppresses their proliferation and immune response [48,49]. Therefore, activation of PD-1/PD-L1 signaling serves as the main mechanism by which a tumor evades antigen-specific immunological response of T cells. Development of PD-1 and PD-L1 inhibitors are becoming major tools in cancer immunotherapy [50].

Activated PSCs also exercise extensive metabolic crosstalk with other cells in PDAC tumor that promotes desmoplasia development, immunosuppression, and cancer progression [22]. The crosstalk between cancer cells and PSC is bidirectional. Cancer cells stimulate via platelet-derived growth factor (PDGF) proliferation of PSCs, and via fibroblast growth factor (FGF-2) and TGF-β1 secretion of collagen type I and III and fibronectin [44,51]. PSCs on the other hand produce periostin, that stimulates cancer cell growth and increase cancer cell resistance to hypoxia [52]. Activated PSCs produce more high molecular weight hyaluronan than quiescent PSCs, which promotes pancreatic cancer cell migration via paracrine signaling [53].

One of the hallmarks of PDAC is the remarkable resistance to chemotherapy that cannot be simply due to low drug delivery (reduced perfusion due to hypovascularity and high turgor pressure due to fibrosis), but also due to a great variety of genetic and metabolic adaptations, and cellular and molecular interactions within TME [54,55]. Some of the interesting mechanisms in gemcitabine resistance include stroma and autocrine factors (see above) that promote EMT, target nucleoside transporters, and cytidine deaminase [54]. Recent report indicates that mast cells, or rather conditioned medium from mast cells, induced drug-resistance in PDAC cell lines to gemcitabine/nab-paclixatel by reducing apoptosis, promoting tumor, and activating TGF−β1 signaling [56]. Furthermore, TGF−β1, tryptase and other pro-inflammatory and immunosuppressive cytokines were increased in the blood of unresponsive patients. In some cancer models, cytokines such as TGF−β1, cause release of ATP, which can affect P2 receptors [5]. In lung cancer cells, TGF−β1 induced ATP release and via activation of the P2X7R it affected actin remodeling and cancer cell migration [57,58]. Whether similar scenario is valid for PDAC remains to be established.

Non-cellular components within TME (e.g., ECM proteins, cytokines, metabolites, acid) support tumor proliferation, cell migration, invasion, and immune system escape [21,23]. One important molecule is the extracellular ATP (eATP). ATP is released from many cells as part of physiological and pathophysiological responses; e.g., cells with high metabolism, such as cancer cells, are proposed to have significant “ATP leak”. Importantly, eATP is a well-recognized “stress” molecule in sterile inflammation [59,60,61,62]. Below we will develop the concept that some steps in the purinergic signaling in pancreas physiology may constitute important players in PDAC development.

## 3. Purinergic Signaling in Exocrine Pancreas—From Physiology to Disease

Purinergic signaling has been studied extensively in pancreatic exocrine and endocrine function [63,64]. Of particular relevance for physiological and pathophysiological processes may be that even in the healthy pancreas there are stores of intracellular ATP in zymogen/secretory granules of pancreatic acini. Here, physiological agonists induce ATP release from granules and about 10 μM ATP was detected in a close vicinity of the plasma membrane [65,66]. Since acini form more than 80–90% of the tissue mass, the ATP reservoir is considerable. Interestingly, pancreatic acini express mRNA for P2X1, P2X4, P2Y_2_, and P2Y_4_ receptors (notably not the P2X7R), but P2 receptors do not seem functional, at least acini often lack the expected Ca^2+^ signaling that should be elicited by stimulation of these P2 receptors; and we proposed that perhaps acini evolved a self-protecting mechanisms [67]. In a physiological situation, acinar ATP is secreted into the duct system, where it via P2Y_2_, P2Y_4_, P2Y_6_, P2Y_11_, P2X4, P2X7 receptors regulates various Cl^−^ and K^+^ ion channels, transporters and Ca^2+^ and also cAMP signaling (TMEM16A/ANO1, CFTR, K_Ca_3.1, K_Ca_1.1, NCX), and thus contributes to epithelial secretion of NaHCO_3_-rich fluid as reviewed [64]. In addition, ducts themselves release ATP via membrane bound channels in response to, for example, mechanical stress and bile acids [68]. Acini and ducts also express ecto-nucleotidases CD39 and CD73 [69,70,71,72]. The resulting product is luminal adenosine that via ductal P1 receptors, in particular A_2A_ and A_2B_, stimulates cystic fibrosis transmembrane conductance regulator Cl^−^ channels (i.e., CFTR), and thus further contributes to pancreatic exocrine function [73,74]. The key role of purinergic signaling in regulation of ion channels/transporters is in fact not only relevant to physiological function, but also to cancer. This because ionic TME and ion channels/transporters are important in regulation of membrane potential, cell volume, cell proliferation, cell cycle, cell migration, acid/base transport, immune suppression etc., and the altered “transportome” is considered as a hallmark of cancer—also of pancreatic cancer [75,76,77].

Pancreatic damage/injury, as may happen in pancreatitis, would cause significant ATP release and activation of purinergic signaling would affect stromal and immune cells. Pancreatitis and inflammation are important factors in pancreatic cancer and a few studies started to consider the role of purinergic signaling in pancreatitis [78,79]. In pancreatic cancer, there are additional sources of ATP—necrotic cells, highly active cancer and stromal cells, which presumably release ATP to the extracellular compartment. Moreover, pancreatic cancer cells have reprogrammed metabolism, including increased glycolysis, greater intracellular ATP production and demand for cellular and intercellular metabolic crosstalk in TME [80]. Presumably, ATP released to TME could contribute to this crosstalk. At least we know that metabolically active PDAC cells lines release significant ATP in response to various stimuli, which may occur in tumor or pre-tumor stages, such as mechanical and osmotic stress, pH changes, and response to bile acids and several releasing mechanisms are involved, including Panx-1 and VNUT [65,66,68,81]. Additionally, at least one PDAC cell line can generate ATP from ADP and UTP by means of adenylate kinase and nucleoside diphosphate kinase [68]. Most importantly, a recent report shows that the intra-tumoral interstitial fluid contains higher level of ATP than the adjacent non-tumor tissue in both humans and mice [82]. One also expects high adenosine in the tumor tissue, as PDAC cells and human tissue have several functional NTPDases (CD39 family) and CD73 [68,83]. 

Turning back, early studies on pancreatic cancer have shown that extracellular nucleotides/-sides have effects on pancreatic cancer cells and cancer. ADP or ATP (<100 μM) produced cell arrest in several PDAC cell lines, including Capan-1 cells and PANC-1 cells [84,85]. Additionally, adenosine was proposed to be important because dipyridamole, which prevents uptake of adenosine, prevented human pancreatic cancer cell-induced hepatic metastasis in nude mice, as well as tumor-induced platelet aggregation in human plasma [86]. Therefore, let us analyze which P2 and P1 receptors, as well as ecto-nucleotidases, are expressed in pancreatic cancer cells and other cells from TME and how they might contribute to PDAC development.

## 4. Purinergic Signaling in Pancreatic Cancer

### 4.1. P2X Receptors

P2X receptors (P2XR) are trimeric ion channels composed of homo- or heteromers and they mediate Na^+^ and Ca^2+^ influx into the cells and K^+^ efflux out of cells. The P2X1-P2X7 receptors are widely distributed in excitable and non-excitable cells. They contribute to cell membrane depolarization by virtue of Na^+^ influx and to modulation of neurotransmission processes that depend on increase in intracellular Ca^2+^ by Ca^2+^ influx [10,11]. In immune cells, activation of P2XR causes loss of intracellular K^+^ and activation of pro-inflammatory pathways [87]. P2XR are widely expressed in epithelia (respiratory, urogenital, kidney, gastrointestinal) and especially in exocrine glands, such as salivary, tear glands and pancreas, where they, by means of Ca^2+^ transients/influx, regulate secretion [64,88]. 

#### 4.1.1. P2X7 Receptors

One of the receptors that has gained most attention in the past years is the P2X7R. It is highly polymorphic with several splice variants (P2X7A-J) and many SNPs, that are associated with a number of pathophysiological processes, like central nervous system diseases, neuropathic pain, osteoporosis, leukemia, obesity, diabetes—all with inflammation as a common ground [89,90,91,92,93]. The receptor responds to physiological supposedly low concentrations of ATP as an ion channel, but as established from in vitro studies, it responds to high micro-to-millimolar ATP concentrations and leads to pore formation allowing permeation of molecules up to 900 Da. Although a crystal structure of the receptor is published [94,95]), there are many questions still unresolved in P2X7R biology, for example, whether the unusually long C-terminal contributes to pore formation, or whether ion channel and pore form simultaneously, which native molecules permeate P2X7R in vivo and how the receptor promotes trophic effects on one hand and causes cell death on the other. Nevertheless, there is a revival of interest in therapeutic targeting of this receptor in inflammation, neurological diseases, and cancer, but there are only a few ongoing clinical trials (Table 1).

P2X7R may be one of the relevant purinergic receptors in various cancers [4,96,97,98]. In general, extracellular ATP can activate P2X7R with different possible outcomes. First, the best established function of the receptor is as a cytolytic pore that occurs with high eATP concentrations and could lead to cancer cell death [4,96]. Second, the P2X7R is a Ca^2+^ channel promoting trophic effects on cell proliferation, cell metabolism, migration, invasion and cellular crosstalk. Therefore, it can also be considered pro-tumorigenic and probably occurs at lower concentrations of eATP (probably activating heteromeric P2X7A–P2X7B) [96,99,100,101,102,103,104]. Third, P2X7R is expressed in tumor immune cells, and it can have both anti- and pro-tumorigenic effects [1,4,97,105,106].

*P2X7R in PDAC cells*: The first study on human pancreatic tissue, reported a tendential increase in P2X7R protein levels in chronic pancreatitis and pancreas cancer compared to normal tissue, although cellular localization was not established [79,107]. Since in the normal pancreas, P2X7R is expressed in ductal cells (see above), it was not surprising to find the receptor in the “normal” human duct cell line HPDE (H6c7) and a high expression in several human PDAC cell lines (PANC-1, BxPC-3, MIA PaCa-2, Capan-1, AsPC-1) with different grades of differentiation and aggressiveness [101]. Functional studies revealed that endogenous/basal and low levels of eATP increased cell proliferation, migration and invasion, while high eATP concentrations induced pore formation and cell death [78,101,108]. Moreover, P2X7R inhibitors and silencing the receptor, reduced cell proliferation and other receptor-mediated effects [102,109]. The P2X7R activates a multitude of downstream events [110]. In rodent pancreas, P2X7R activates ERK1/2 signaling via the N-terminus. In PDAC cells, ERK1/2 activation and promotes cancer cell cycle progression [103]. In other studies, it was proposed that statins, allegedly working via P2X7R, inhibited PDAC progression via P3IK/Akt signaling [109,111]. However, statins lower cholesterol and caveolins, and since caveolins associate with P2X7R [110], statin effects are most likely indirect and dependent on lipid metabolism and lipid composition of the plasma membrane of cancer cells.

*P2X7R in TME cells*: Pancreatitis is an inflammatory disease and an important risk for PDAC, and early studies/immunocytochemistry indicated expression of P2X7R in several cell types in the inflamed tissue—PSCs, leucocytes and presumably macrophages [79,112]. In PDAC, PSCs are important cells in TME and several studies show that these cells from rodent pancreas respond with Ca^2+^ transients to nucleotides (ATP, UTP) and express several P2Y and P2X7 receptors [78,108,113]. More detailed study on isolated mouse PSCs showed that they express several isoforms of P2X7R (A–D, K), and stimulation of activated cells in culture with ATP and BzATP in low concentrations (<100 μM) increases cell proliferation, while high concentrations (>100 μM to mM) induced cell death [113]. PSC isolated from Pfizer KO mice on C57BL/6JBom background, which expressed B, C and truncated A and K isoforms, were resistant to cell death, but they also proliferated more slowly. The actual number of PSC in the pancreas was about half of that in wild-type animals, indicating that A and/or K isoforms are important for optimal regulation of cell proliferation [113]. Another function of P2X7R in PSCs is to attract PDAC cells by releasing yet to be identified chemoattractant. This was shown in a study where PSCs isolated from the P2X7R KO mice could not support migration of PDAC cells (PancTu-Luc) as wild PSCs did [102,113].

Purinergic signaling is also emerging as a communication pathway between CSCs and TME components. One interesting study shows that tumor-associated macrophages in PDAC microenvironment secrete immunomodulatory cationic peptide 18/LL-37 that via P2X7R expressed on CSCs (CD133^+^) increased pluripotency-activated genes, self-renewal, invasion, and tumorigenicity in PDAC [114]. P2X7R inhibition with KN-67 significantly reduced in vitro CSC colony formation and invasion, and in vivo tumor development and liver metastasis. As reviewed recently, several studies show that other non-nucleotide agonists, including antimicrobial cathelicidin LL-37 peptide, activate P2X7R and pore formation, though the underlying mechanisms are not clear [115]. 

The P2X7R is expressed in virtually all innate and adaptive immune cells, and it can modulate both pro-inflammatory and anti-inflammatory responses, thus in cancer, contributing to tumor-promoting and/or tumor suppressing responses [96,106,116]. The outcome will depend on the tumor-TME interaction and the characteristics of the resident and infiltrating inflammatory cells, as demonstrated in other types of cancer. For example, in animal models of colon carcinoma, melanoma and leukemia, lack of host P2X7R dampens anti-cancer response and promotes tumor growth, due to reduction of a number of CD8^+^ (cancer killer T cells) and increased T_reg_ overexpressing fitness markers [116]. However, the PDAC harbors highly immunosuppressive TME (see above) and the oncogenic role of P2X7R in cancer cells and fibrogenic PSCs may prevail. 

*P2X7R in in vivo models of pancreatic cancer*: The role of P2X7R in PDAC has been investigated in two in vivo studies performed on different mice models. In one study, an orthotopic xenograft model of PDAC was used. Human PDAC cells, PancTu-1 implanted into the pancreas of male nude mice developed rather aggressive tumors at 10 days and they were treated with intraperitoneal injection of P2X7R inhibitor AZ1060612 [102]. The P2X7R was highly expressed in the tumor niche—cancer cells, PSCs, PanINs, and perhaps unexpectedly in ADMs. The AZ1060612 treated group showed a significant reduction in tumor activity/bioluminescence compared to control even if, after sacrifice, the tumor masses of two groups were not significantly different. However, fibrosis, collagen deposition as well as PSC number was significantly reduced in the treated animals. The key role of P2X7R in PSCs was verified in in vitro experiments [102,113], as described above (P2X7R in TME cells).

Another study [117] used mouse model of pancreatic cancer (p48^cre/+^-LSL-Kras^G12D/+^) bred on C57BL/6 genetic background, a mouse strain which carries P451L loss of function mutation and lower expression of the receptor on T cells [118]. Transcriptomic analysis of *KRAS* tumors showed that the key P2X7R inflammasome components—IL-1β and caspase-1 and other inflammatory markers were highly expressed compared to normal pancreas [117]. The two inhibitors, AZ10606120 or A438079 (50 ppm), were given orally for 38 weeks and the total pancreas weight taken as an indication of tumor size and histopathology was used to estimate PDAC incidence. The percentage of carcinoma increased in drug-treated male mice, while it decreased in drug-treated female mice. Close analysis of inflammasome markers in male pancreas revealed no clear correlations with the two inhibitors.

Although there is evidence that the P2X7R supports oncogenesis, fibrosis and inflammation in pancreas, the role of this receptor in the immune environment/components is not clear. Therefore, future studies should clarify whether an intact immune system would have contributed to PDAC progression in treated animals, whether there are differences in males vs females with respect to P2X7R function in cancer, as they are in the overall pancreatic exocrine function [117,119], and whether genetic variants in rodent and human P2X7R [89,90,110] contribute to PDAC development. Moreover, the role of TGF−β1 in autocrine stimulation of P2X7R in PDAC remains to be clarified. So far, there are a handful of exploratory clinical trials on P2X7R in other cancers but not PDAC (Table 1).

#### 4.1.2. Other P2X Receptors

Several P2X receptors (P2X1, P2X2, P2X4, P2X5, P2X6, P2X7) are expressed in human PDAC cell lines [101,120]. Interestingly, also P2X5R gene is highly upregulated in human pancreatic tumors compared to normal pancreas samples, as determined in gene analysis of ion channels/transport proteins “Transportome” [121], but information about contribution of this receptor to PDAC behavior in vitro or in vivo is yet not available. However, some studies on P2X5R in other cancer forms are published. P2X5 receptors have been identified in squamous cell carcinomas of the skin and prostate cancers and different grades of papillary urothelial carcinoma [122,123,124]. For more complete reviews on P2XR and other cancers refer to [4,125].

### 4.2. P2Y Receptors

P2Y receptors P2Y_2_, P2Y_4_, and P2Y_6_ are expressed in pancreatic ducts and the P2Y_2_ in particular is very important in regulation of Cl^−^ channels (TMEM16A, CFTR) and K^+^ channels (K_Ca_3.1 and K_Ca_1.1), which are essential for pancreatic duct secretion (see above). Several PDAC cell lines (PANC-1, CFPAC-1, MIA PaCa-2, BxPC-3, AsPC-1 and Capan-1) express P2Y_1_, P2Y_2_, P2Y_4_, P2Y_6_, P2Y_11_, P2Y_12_, P2Y_13_ and P2Y_14_ receptors on mRNA level and protein level [101,120,126]. When some of these receptors are stimulated by ATP or UTP, they also regulate Cl^−^ and K^+^ channels (TMEM16A, K_Ca_3.1) that are over-expressed in PDAC and thus may contribute to its progression [127,128]. Further functional studies on specific receptors on PDAC in vitro and in vivo models are given below.

#### 4.2.1. P2Y_2_ Receptors

The first seminal study on pancreatic tissue from patients with chronic pancreatitis and pancreas cancer was by Kunzli and co-workers [107]. The study shows that the mRNA and protein levels of P2Y_2_ and triphosphate diphophohydrolases (NTPDase-1 and -2) were highly expressed in pancreatic tissue of patients suffering from pancreatic cancer compared to normal pancreas samples [107]. The high expression of P2Y_2_ was associated with poor prognosis, whereas the high expression of NTPDases in malignant tissue indicates progression of tumor development induced by P2Y_2_ in PDAC. Expression of P2Y_1_ and P2Y_6_ in normal and diseased pancreas were similar [107]. A recent study on a large number of PDAC samples demonstrated the upregulation of P2Y_2_ receptor and associated poor prognosis in patients [82]. The same study also reports higher P2Y_2_ protein expression in PanINs and PDAC tissues compared with normal acini in genetically engineered mouse model of LSL-Kras^G12D/+^; LSL-Trp53^R172H/+^; Pdx1-Cre (KPC). The P2Y_2_ receptor is also highly expressed in some PDAC cell lines compared to normal cells [82,101].

On the cellular level, similar to P2X7R, the P2Y_2_ receptors are involved in cell growth and differentiation, cell migration, inflammation, and fibrosis and can have diverse roles in different cancers [12,17,116,129]. In PDAC cells PANC-1, the role of P2Y_2_ receptor was demonstrated in two ways: UTP and P2Y_2_ agonist MRS2768 increased cell proliferation; siRNA and P2Y inhibitor suramin decreased cell proliferation [126]. Further, the data indicate that the P2Y_2_ receptor effects were dependent on the key signal mediators—phospholipase C, inositol 1,4,5 triphosphate (IP3), protein kinase C, and PI3K/Akt signaling mediated by PKC, SFK, and CaM kinase II [126]. 

In another study, application of UTP, ATP, as well as genetic and pharmacological inhibition of P2Y_2_R (shRNA of P2Y_2_ and inhibitor AR-C118925XX) increased cell proliferation and glycolysis in several PDAC cell lines [82]. The study also indicated that enhanced glycolysis was due to P2Y_2_ activation of PI3K/Akt-mTOR signaling by crosstalk with PDGFRβ mediated by Yes1, following higher expression of c-Myc and HIF1α. No significant activation of EGFR was observed in this study, in contrast to studies on glioblastoma and salivary gland cells [130,131]. Thus, ATP promotes pancreatic cancer cell progression by reprogramming the metabolism. Genetic and pharmacological inhibition of P2Y_2_R reduced tumor growth in subcutaneous and orthotopic xenograft models and also delayed tumor progression in inflammation-driven PDAC model. Most importantly, AR-C118925XX had a synergistic effect when combined with gemcitabine on tumor suppression and survival rate in xenograft PDAC mice [82]. 

P2Y_2_ (and other P2Y and P2X receptors) are also expressed in activated PSCs and show UTP-induced Ca^2+^ signaling [108], but final effects on cell behavior are not known. Since P2Y_2_R is an important receptor in inflammation and fibrosis, for example in the liver [129,132], and considering that endothelial P2Y_2_R (activated by platelet-derived ATP) promotes leakage and tumor cells trans-endothelial migration and metastasis [133], further investigation of the role of this receptor in these TME processes in PDAC will be valuable.

#### 4.2.2. P2Y_12_ Receptors

One of the complications of PDAC can be venous thromboembolism, which has complex pathophysiological mechanisms that are only partly understood and still discussed [134,135]. Platelets are the key player in hemostasis and thrombosis, they express the ADP-activated P2Y_12_ receptor, which is targeted by Clopidogrel to reduce the risk of stroke and heart disease. There is renewed interest in signaling between platelets and cancer cells, which promotes transition of cancer cells from primary tumors to invasive mesenchymal phenotype, enhancement of metastasis and induction of chemoresistance in pancreatic cancer [136,137,138]. In pancreatic cancer mouse models, Clopidogrel restored hemostasis, decreased tumor size and prevented development of metastases [139]. In recent studies, Elaskalani and co-workers asked the question whether there is platelet-pancreatic cancer cell crosstalk and whether pancreatic cancer cells also express P2Y_12_R [140,141]. They found that the P2Y_12_R is over-expressed in several PDAC cell lines compared to control pancreatic cell line (hTERT-HPNE) [140]. Platelet releasate (includes ATP and ADP from granules) activated Akt and ERK signaling and stimulated expression of markers of EMT (e.g., SLUG) and promoted proliferation of PDAC cells. It also downregulated expression of equilibrative nucleoside transporter (hENT1) and cytidine deaminase, which are markers of gemcitabine resistance in pancreatic cancer [141]. P2Y_12_ R-siRNA knock down and other pharmacological inhibitors prevented EGF-P2Y_12_R crosstalk, Akt activation, expression of SLUG and ZEB1 and cancer cell proliferation. Importantly, Ticagrelor, a P2Y_12_R inhibitor in clinical use, enhanced effects of chemotherapeutic drugs on cell viability and apoptosis in vitro [141]. However, in two in vivo models (nude mice with orthotopic BxPC-3 tumor and syngeneic model of C57BL6/J mice with MT4-2D), Ticagrelor had no significant effect on tumor volume on its own, but reduced it when given together with gemcitabine [141]. Nevertheless, it will be interesting to follow the outcome of the clinical trials where Clopidogrel is given in addition to chemotherapy in PDAC patients (Table 1).

#### 4.2.3. Other P2Y Receptors 

ADP- activated P2Y_1_ and UDP-activated P2Y_6_ receptors are expressed in PANC-1 cells and other PDAC cell lines [101,120,142]. Specific agonists/inhibitors show that stimulation of these two receptors increases the proliferation of PANC-1 cells [142]. In addition, P2Y_1_R may contribute to tissue development in glands, e.g., salivary glands [88], and in pancreas, recent studies show that P2Y_1_R is an additional marker on pancreatic and duodenal homeobox 1 (PDX1) progenitor-like cells from human pancreatic tissue [143].

The P2Y_11_ receptor is expressed in several PDAC cell lines [101], and based on the effect of inhibitor NF157, it was reported that this receptor is a co-regulator of cell migration in PDAC cell lines BxPC-3 and Capan-2 [144]. The prime regulator of cell migration is the protease activated receptor 2 (PAR-2) that activates EGFR-Src-Rac-p38/MEK/ERK1/2 signaling pathway in the presence of ATP [144].

### 4.3. Adenosine Receptors

The extracellular fluid of solid carcinomas contains adenosine predominantly produced by sequential hydrolysis of eATP (see above). The four types of adenosine receptors are expressed in rodent pancreatic ducts and PDAC cell lines PANC-1, CFPAC-1, and Capan-1 (on the mRNA level expression is A_2A_ and A_2B_ >> A_3_ and A_1_) [73,145]. The A_2A_ receptor stimulates Cl^−^ channels (CFTR) in native pancreatic ducts of rat, mouse and guinea pig and contributes to exocrine secretion [74]. In PDAC cells it is most likely that the A_2B_ receptor stimulates Cl^−^ ion channels (CFTR) [73], but it is not clear whether this A_2B_-Cl^−^ channel axes contributes to tumorigenicity. On the mRNA level, A_2B_ showed the highest expression in the PDAC cell lines [145]. In one study on two PDAC cell lines (and other cancer cell lines) it was shown that stimulation of A_2B_ receptor reduced cell–cell contact and increased cell scattering, implicating its role in metastatic spreading [146]. Gene expression analysis of pancreatic cancer tissues based on TCGA and GTEx databases (The Cancer Genome Atlas and Genotype-Tissue Expression projects) showed that both *ADORA2A* and *ADORA2B* were significantly upregulated [147]. Interestingly, analysis of TCGA database, indicates that the high expression of *ADORA2B* is associated with poor survival, while high expression of *ADORA2A* is associated with significantly better survival [74]. In a number of other cancers and cancer models, e.g., prostate, breast and colon, the A_2B_ receptor is upregulated and associated with poor prognosis and A_2B_ inhibitors are anti-proliferative and may increase sensitivity to chemotherapeutic drugs [13,148,149,150].

Adenosine is immunosuppressive and acts on various immune cells that predominantly express the A_2A_ receptor [6]. For example, A_2A_R is upregulated during inflammation on effector T cells and inhibits T cell proliferation, cytotoxic activity and cytokine production, while activation in T_reg_ promotes their expansion and immunosuppressive activity. A_2A_R activation on natural killer cells inhibits their maturation, proliferation, activation, and secretion of cytotoxic cytokines [6,151]. Nevertheless, with respect to PDAC, finer resolution of A_2A_ and A_2B_ expression in TME, i.e., in cancer vs immune cells, will be required to understand survival curves (see above) and determine whether A_2A_R is immunosuppressive and will constitute a successful target in ongoing clinical trials (Table 1).

Adenosine A_3_ receptors are the most studied adenosine receptors in several cancers, and opposing effects can be found ranging from anti-tumoral effects to those promoting metastasis [3]. A_3_ receptor is upregulated in breast and colon tumor tissues compared to the adjacent normal tissue, but the number of samples from pancreas was limited to reach any conclusion in one study [152], but gene expression analysis of pancreatic cancer tissues based on TCGA and GTEx databases shows that also *ADORA3* is upregulated compared to normal pancreas [147]. Recent interesting papers [153,154] show that adenosine, A_3_R, and mast cell—cancer cell interactions may constitute an important part in TME. A_3_R is expressed in mast cells that are activated by direct contact with plasma membranes of PDAC cells or extracellular vesicles derived therefrom. These express CD73 (and presumably CD39) and from whatever source of ATP originates from, they generate adenosine. A_3_R signaling via ERK1/2 stimulation leads to upregulation of genes for IL-6, IL-8, VEGF, and amphiregulin (ligand for EGFR) in mast cells and thus shift towards pro-tumorigenic profile [153,154].

### 4.4. CD39 and CD73

Sequential activity of CD39 (ATP/ADP to AMP) and CD73 (AMP to adenosine) are largely responsible for generation of adenosine in TME. Kunzli and co-workers [107] have nicely shown that CD39 enzymes, both on mRNA and protein level, were upregulated in pancreatic tissues from patients with pancreatic cancer (and also chronic pancreatitis), and higher expression correlated with better long-term survival of cancer patients after tumor resection. Using immunocytochemistry, the enzymes were localized to vasculature and stroma, and later to pancreatic stellate cells [107,112], pancreatic acini and ducts and secreted microvesicles [68,70,71]. Another ecto-nucleotidase/multifunctional protein, CD73, is expressed in apical/luminal membrane of pancreatic ducts and PDAC cell lines and it contributes to adenosine generation [68,71]. Interestingly, localization of CD73 in normal duct is apical/luminal, but becomes aberrant and cytoplasmic as PanINs progress to PDAC (but not other pancreatic cancers such as acinar cell carcinoma) and it correlates with loss of E-Cadherin [72].

Several reports show that CD73, on both mRNA and protein levels, is upregulated in pancreatic cancer and PDAC cell lines [72,83,147,155] and in mouse organoid isografts [156]. In particular, the recent bioinformatics study of several cohort databases reveals some interesting correlations [147]. Upon stratification, the study showed that patients with the basal rather than the classical cancer subtype had poorer survival rate with higher CD73 expression, and high CD73 expression was associated with lower DNA methylation. Interestingly, CD73 expression level was negatively correlated with infiltrating levels of CD8^+^ T cells and γδ+ T cells, and positively correlated with PD-L1 expression, both promoting tumor immune escape. The authors propose that CD73 might be a promising biomarker as well as a co-target together with anti-PD-1/PD-L1.

Recent reports indicate that CD73 has other non-enzymatic activities, also in pancreatic cancer [83,157]. Thus, CD73 knockdown inhibited cell growth and induced G1 phase arrest via Akt/ERK/cyclin D signaling and TNF receptor2 was involved in this CD73-induced Akt/ERK signaling. In addition, since overexpression of miR-30a-5p targets CD73 and increases sensitivity of pancreas cancer to gemcitabine, the authors propose a model for mechanistic interaction between miR-30a-5p, CD73, and TNFR2, supposedly on pancreatic cancer cells and independent of adenosine generation in this scenario [83].

CD39 and CD73 are expressed in a number of immune cells [158] and a recent study shows that perhaps unusual conditions may prevail in PDAC. Specialized pancreatic tumor-infiltrating dendritic cells (CD11b^+^CD103^−^) induce tumor-promoting Tr1-like phenotype of CD4^+^ T cells that, among others, show high expression of CD39 and CD73, and promote tolerance to PDAC and resistance to immunotherapy [159].

In addition to chemoresistance, pancreatic cancer is also exceptionally resistant to radiotherapy. One study indicates that CD73 may be involved as follows. Irradiation selected pancreatic cancer cells (MIA PaCa-2) had inhibited apoptosis and higher degree of EMT plasticity and proteomic, and functional analysis revealed that one of the most upregulated proteins was CD73 [160]. CD73 confers acquired radioresistance by inactivating pro-apoptotic protein BAD and maintaining cells in a mesenchymal state. Whether this is a chain of reactions set into action due to irradiation damage, ATP release and enzymatic hydrolysis by CD39 and CD73, or non-enzymatic effects of CD73, is not yet clear. Nevertheless, CD73 is proposed as a promising immunotherapy target in cancer, including PDAC [6,151].

The above part of purinergic signaling, so called adenosinergic pathway that includes adenosine producing enzymes/multifunctional proteins and adenosine receptors, has been successfully exploited as potential approach to overcome tumor-induced immunosuppression [7]. Namely, co-inhibition of CD73 and A_2A_R improves anti-tumor response in several cancer models [151,157]. An overview of the clinical trials to target adenosinergic pathway, especially CD73 and CD39, in PDAC together with other co-therapy and irradiation is given in the Table 1 together with other purinergic signaling targets. A recent review summarizes involvement of adenosinergic molecules in other human cancers, and ongoing clinical trials with inhibitors of these pathways together with combination therapies [7].

## 5. Conclusions and Future Perspectives

Purinergic signaling is an integral part of exocrine pancreas physiology. It is now accepted that pancreatic acini release ATP, pancreatic acini and ducts hydrolyze ATP to adenosine and both ATP, ADP and adenosine regulate ion channels and thus pancreatic duct secretion. It seems that some of the very same receptors and enzymes in pancreas physiology are also upregulated in pancreatic cancer, not only in cancer cells, but also in stromal cells and possibly immune cells, as depicted in Figure 1. However, purinergic signaling components in immune cells in highly suppressive PDAC microenvironment have not been explored in detail yet. The P2X7 (and P2Y_2_) receptors seem particularly promising targets to curb inflammation, cancer cell proliferation and pancreatic stellate cell activity, though clinical trials for any P2X7R (and P2Y_2_ R) related diseases are still few. In contrast, the ADP-stimulated P2Y_12_ receptor has not been explored yet as epithelial regulator, though it seems an interesting double-edged sword for targeting cancer cells and platelets for decreasing cancer progression and thromboembolism in cancer patients. Drugs targeting CD39, CD73, and A_2A_ receptor have been taken swiftly into clinical trials for immune therapy, though these proteins are not only confined to immune cells, but also to epithelial cells, and thus caution needs to be exercised. Along the same lines, one can predict that inhibition of ATP degradation would lead to increase in extracellular ATP and thus activation of low affinity P2X7 receptors. If it would be possible to stratify the patients to those demonstrating pro-inflammatory phenotypes (early steps in purinergic signaling) and those demonstrating predominantly immunosuppressive phenotypes (adenosinergic steps), more precise clinical trials could be designed. This would require a large repertoire of biomarkers including soluble and exosome-bound factors such as CD73/CD39, “soluble” P2X7R, inflammatory and immunosuppressive factors, immune cell analysis, molecular profiling, etc. In addition, since purinergic signaling is important for the digestive function of pancreas, also the nutritional status should be considered. Therefore, we need to know more about the integrative role of purinergic signaling in pancreas in total, as pancreas cancer patients with good pancreatic function (exocrine and endocrine) would have a better chance of surviving pancreatic cancer.

## Figures and Tables

**Figure 1 ijms-21-08781-f001:**
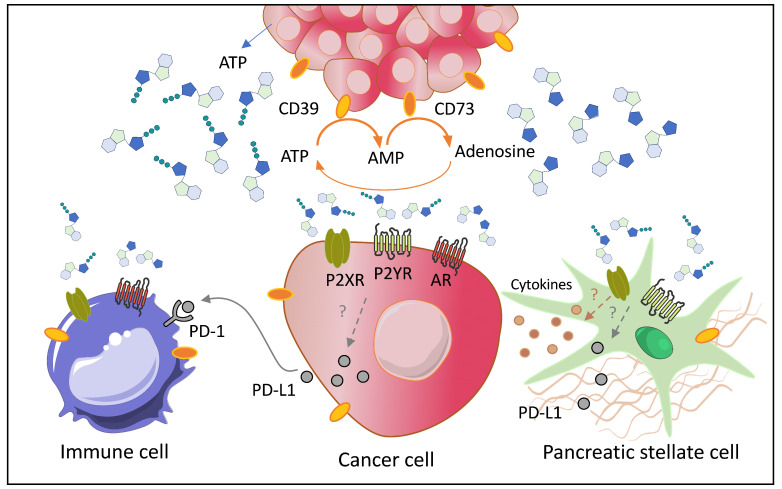
Schematic diagram of pancreatic tumor (upper part) and the key cells in the tumor microenvironment (TME) (detailed in the lower part)—cancer cells, pancreatic stellate cells (PSCs), and immune cells and key proteins in the purinergic network that are highly expressed in pancreatic ductal adenocarcinoma (PDAC). Cancer cells, and epithelial precursors (duct and acinar cells, see the abstract figure), have both several mechanisms to release ATP, which is then hydrolyzed via CD39 and CD73 to adenosine. In pancreas/PDAC, extracellular ATP stimulates P2 receptors, such as the P2X7, P2Y_2_, and P2Y_12_, while extracellular adenosine stimulates A_2A_ and A_2B_ receptors. PSCs secrete cytokines and extracellular matrix (ECM) proteins, express P2X7 and several P2Y receptors (see text). Immune cells express various P2X, P2Y, and A receptors (schematic). The model also shows programmed cells death protein-1 ligand (PD-L1) that is secreted by cancer cells and PSCs and binds to PD-1 receptor on T cells (and other immune cells). This constitutes the signaling network in TME by which the tumor may evade antigen-specific immunological response. Whether purinergic signaling regulates cytokine release or secretion of PD-L1 in PDAC is not known (indicated by?). Drugs or antibodies targeting one or more components of the purinergic/adenosinergic/PD-1 signaling are in some clinical trials for PDAC (Table 1).

**Table 1 ijms-21-08781-t001:** Clinical trials (from NIH and EU registers) testing the potential use of purinergic targets in pancreatic and other cancers. Purpose of the studies varies from feasibility to safety, tolerability, pharmacokinetics, immunogenicity, and anti-tumor activity.

Target	Drug +/− Combination Therapy or Outcome	Sponsor/Company	Cancer	Identifier	Study Phase
P2X7	Biomarker, histopathologic response, analysis of inflammasome and polymorphisms, ointment with Ab against non-functional P2X7R	various	Inflammation, ovarian cancer, colon cancer, basal cell carcinoma, endometrial cancer, breast cancer	Ovarian Colon cancer NCT04122937 Breast cancer NCT01440413	Various stages
P2Y12	Clopidogrel (P2Y12 antagonist) and chemotherapy	Assistance Publique-Hôpitaux de Paris	Treatment of locally advanced or metastatic pancreatic cancer	NCT02404363	Phase III No results poster
A2A	NIR178 in (A2A antagonist) in combination with PDR001 (anti-PD-1 Ab)	Novartis Pharmaceuticals	Patients with solid tumors (including pancreatic cancer) and Non-Hodgkin Lymphoma	NCT03207867	Phase II
CD39	TTX-030 (anti-CD39 Ab) in combination with standard chemo- or immunotherapy	Tizona Therapeutics	Patients with advanced cancers	NCT04306900	Phase I/Ib
	SRF617(anti-CD39 Ab)	Surface Oncology	Patients with advanced solid tumors to improve immune response	NCT04336098	Phase I
	TTX-030 (anti-CD39 Ab) +/− anti-PD1 immunotherapy	Tizona Therapeutics	Patients with advanced cancers	NCT03884556	Phase I
CD73	AB680 (CD73 inhibitor) +/− AB122 (PD-1), nab-paclitaxel and gemcitabine	Arcus Bioscience	Advanced pancreatic cancer	NTC04104672	Phase I
	LY3475070 (CD73 inhibitor) +/− Pembrolizumab	Eli Lilly and Company	Patients with advanced solid malignancies including pancreatic cancer	NCT04148937	Phase I
	MEDI9447 (oleclumab, anti-CF73 Ab) and MEDI4736 (durvalumab, anti PD-L1–PD-1 Ab)	MedImmune LLC	Patients with solid tumors including pancreatic cancer	NCT02503774	Phase I
	BMS-986179 (anti-CD73 Ab) + Nivolumab (BMS- 936558, anti PD-1 Ab)	Bristol-Myers Squibb International Corporation	Patients with advanced tumors	EudraCT 2016-000603-91	Phase I/IIa
PD1 and CD73	Durvalumab (MEDI4736, anti-PD-L1 Ab) + Oleclumab (MEDI9447, anti-CD73)i	University Health Network, Toronto and Astra Zeneca	Patients with PDAC and other cancers	NCT04262388	Phase II
CD73 +/− A2A	NZV930 (anti-CD73 Ab) +/− PDR001 (anti-PD-1 Ab) +/− NIR178 (A2A antagonist)	Novartis Pharmaceuticals	Patients with advanced cancers including PDAC	NCT03549000	Phase I/Ib
CD73 +/− A2A	CPI-006 (anti-CD73 Ab) +/− ciforadenant (oral A2A inhibitor) +/− pembrolizumab (anti-PD1 Ab)	Corvus Pharmaceuticals	Patients with selected advanced cancers including pancreatic cancer	NCT03454451	Phase I/Ib

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
