# Peer review of "Purinergic Signaling in Pancreas—From Physiology to Therapeutic Strategies in Pancreatic Cancer"

_ijms, 2020, doi:10.3390/ijms21228781_

Round 1
Reviewer 1 Report
The manuscript is an interesting and complete review of the argument, which includes knowledges about the involvement of different components of the purinergic signalling in pancreas physiology (a topic already addressed from these authors in previous publications) and in pancreatic cancers, especially pancreatic ductal carcinoma (PDAC).
The manuscript is clear and well written and very few corrections or modifications are needed
Line 31: “It is predicated” should be changed to: “It is predicted”
Line 95: “The cellular origin of cancer cells” should be changed to: “The cellular origin of pancreatic cancer cells”
Line 218: “..allowing permeation of molecules up to 900 kDa” should be changed to “..allowing permeation of molecules up to 900 Da”
- the paragraph from line 212 to line 224 is all dedicated to the P2X7 receptor. Therefore, it should be moved to the 4.1.1 paragraph that describes the P2X7 receptors.
Line 252: “pancreatic stellate cells” should be changed to “PSCs”
Lines 287-288: “The key role of P2X7R in PSCs was verified in vitro experiments“ should be changed to “The key role of P2X7R in PSCs was verified in in vitro experiments“. A brief description of these experiments (PMID 27513892, 23284663) could be added
Line 295: “It seems … “ does not sound scientifically acceptable. The main result of the cited reference (PMID 29228654) is that the two P2X7R inhibitors did not show any chemo-preventive effect against pancreatic cancer, independently of the animal gender
Line 443: as regard CD73 expression in pancreatic cancer, a very recent paper should be cited (PMID 32643277).
Line 505: “PD-L1” should be changed to “PD-L1”
The figure is too schematic and some details are missing. For example: 1) it is not clear what kind of cells express CD39 and CD73 at the top of the figure, 2) the role of pancreatic stellate cells can not be deduced from the figure, 3) the relationship between the purinergic signalling and PD-1/PD-L1 is not clear from the picture.
Author Response
Dear Reviewer 1,
Thank you for your kind comments and thorough revision of the Ms.
We have corrected all small typos and errors.
Regarding the paragraph on the P2X7R, it has now been moved to the section dedicated to it.
Regarding the role of P2X7R in in vitro experiments (our own) - they are described in the relevant section, expanded slightly and referred to at the end of in vivo part.
Regarding the cited references of Mohammed et al - I agree "it seems" is not very scientific and therefore I removed it. The conclusions presented in their abstract are not precise. In the results (Table 1 and 2) and page 97826 they show that the inhibitors increased the PDAC incidence in males but reduced in females (no statistics is given though). The same is regarding carcinoma percentage, though there are some dose –differences. Expression analysis of predictive markers is then only carried out on male samples. So, I kept the comments on this paper to a minimum.
Regarding the recent paper (PMID 32643277) - thank you for pointing this out. Chen et al paper very good and includes some very good points also for inclusion in other places, which we have done now.
Regarding the figure - We have modified the figure and expanded the figure legend. We only focus on what is known about the purinergic signaling in PDAC and indicate potential but unexplored interaction with other signalling networks with ?
Reviewer 2 Report
This is a good review and balanced assessment of the status of the network of purinergic and adenosine receptors from an authority in the field.
The article highlights important data that might have been overlooked when promulgating the clinical value of P2X7 receptor, P2Y2 and P2Y12 receptors, as well as A2 receptors and ecto-nucleotidases CD39 and CD73 and related studies.
I would only suggest to slightly deeper the discussion regarding tumour microenvironment (TAM) and its role in PDAC drug resistance, in light of some recent pieces of evidence regarding TAM, TGFbeta pathway and purinergic and adenosine metabolism (PMID: 30866547). For instance, it has been demonstrated that treatment with TGF-β1 elicits ATP release from cancer cells, thus activating P2 receptors. Actin remodeling and cell migration induced by TGF-β1 required the expression and autocrine stimulation of P2X7R, since these processes were suppressed after P2X7R knock-down or pharmacological inhibition.
The authors could provide a little more consideration of TME directed stratifications in clinical trial design and enrollments. The underlying message here is that more precision and individualized approaches need to be tested in well designed clinical trials – a challenge, but I would be interested in their perspective of how this might be done.
Author Response
Dear Reviewer 2,
Thank you for your interesting input.
Regarding TME and the drug resistance - I have added a small section on drug resistance in PDAC, which is a huge area or research and possibilities. As suggested I included the suggested paper on the role of mast cells and TGFb1, and refer to a review and a couple of papers on interaction with and purinergic signaling networks as known for the lung cancer models. We have tried to focus strictly on PDAC in this Ms, and we leave the more in depth discussion on cancers in general to other reviews we refer to.
Regarding the stratification and clinical studies - Difficult. I have included a small section in the conclusion paragraph. However, this is very speculative and one needs to consider many more factors than purinergic signalling, which we, as non-clinicians, do not have competencies in.